# Wearable Robot Design Optimization Using Closed-Form Human–Robot Dynamic Interaction Model

**DOI:** 10.3390/s24134081

**Published:** 2024-06-23

**Authors:** Erfan Shahabpoor, Bethany Gray, Andrew Plummer

**Affiliations:** 1Department of Architecture and Civil Engineering, University of Bath, Claverton Down, Bath BA2 7AY, UK; 2Department of Mechanical Engineering, University of Bath, Claverton Down, Bath BA2 7AY, UK

**Keywords:** exoskeleton, orthosis, virtual prototyping, human model-in-the-loop, musculoskeletal simulation, optimization

## Abstract

Wearable robots are emerging as a viable and effective solution for assisting and enabling people who suffer from balance and mobility disorders. Virtual prototyping is a powerful tool to design robots, preventing the costly iterative physical prototyping and testing. Design of wearable robots through modelling, however, often involves computationally expensive and error-prone multi-body simulations wrapped in an optimization framework to simulate human–robot–environment interactions. This paper proposes a framework to make the human–robot link segment system statically determinate, allowing for the closed-form inverse dynamics formulation of the link–segment model to be solved directly in order to simulate human–robot dynamic interactions. The paper also uses a technique developed by the authors to estimate the walking ground reactions from reference kinematic data, avoiding the need to measure them. The proposed framework is (a) computationally efficient and (b) transparent and easy to interpret, and (c) eliminates the need for optimization, detailed musculoskeletal modelling and measuring ground reaction forces for normal walking simulations. It is used to optimise the position of hip and ankle joints and the actuator torque–velocity requirements for a seven segments of a lower-limb wearable robot that is attached to the user at the shoes and pelvis. Gait measurements were carried out on six healthy subjects, and the data were used for design optimization and validation. The new technique promises to offer a significant advance in the way in which wearable robots can be designed.

## 1. Introduction

Wearable robots have a huge potential to assist users with movement and balance disorders and to restore their ability to perform activities of daily living. Virtual prototyping of these robots using modelling and simulation tools allows for kinematic synthesis, topological selection, ergonomic validation and morphological, functional and dynamic optimization of the robot, preventing costly iterative physical prototyping and testing. Design of such devices through modelling, however, requires realistic simulation of human–robot interactions which is challenging due to the inherent variability and unpredictability of human movements, adaptations and reactions.

The past decade has witnessed significant growth in use of multibody, data-driven and physics-based modelling tools for design of wearable robots, to simulate human–robot interactions, to optimize design parameters, and to estimate the effects of assistance and inertia of the robot on the users [1]. Sergi et al. [2] and Accoto et al. [3] used a simplified link–segment model of one leg to simulate interaction of a wearable lower-limb orthotic with an active hip and knee. Ground reaction forces (GRFs) and the inertia of the robot were not considered in the simulation. The measured anthropometry and hip and knee kinematics and kinetics during walking by a standard/generic test subject were used as input. The model was then wrapped in a non-linear optimization framework to optimize the morphology of the robot and the robot joint torques.

Ferrati et al. [4] combined the model of a lower limb exoskeleton with active knee and hip joints with a full-body musculoskeletal model of a human in OpenSim. Generic walking kinematics were used as inputs and GRFs were not considered in the simulations. The human lower limb muscle forces were set to zero to emulate paralysis. The required torques of the robot actuators were found using the OpenSim Computed Muscle Control tool (combination of proportional–derivative control and static optimization) to reproduce human kinematics. Agarwal et al. [5] has similarly proposed a framework based on an index finger exoskeleton model attached to an OpenSim hand model. Measured kinematics and contact forces for a grasping task were used as input. Peak muscle forces were capped to simulate disability, and joint torques in the robot were calculated using the OpenSim Computed Muscle Control tool. The framework was wrapped in an optimization process to find optimum design values for the robot.

Galinski et al. [6] used the Robotran multibody simulation package within a multi-objective genetic algorithm optimization to simulate a shoulder complex and to analyze the required torques at two actuators of the robot. Shourijeh et al. [7] used the Anybody multibody simulation package to simulate interactions of a passive lower-back exoskeleton and the user during a box lifting task. GRFs and kinematics of the human were estimated using Anybody tools, assuming a sixth-order polynomial function for box motion. Robot inertia and dynamics were not considered in the simulations and assistive torques were only used in the model as functions of joint angles.

Kruif et al. [8] proposed a simulation architecture in which the interaction of an elbow-articulated exoskeleton with its user was modelled using musculoskeletal models in OpenSim, and the control algorithm and human response model were created in Matlab. Fournier et al. [9] simulated the interactions of an ARKE lower-limb exoskeleton from Bionik Laboratories with a human user using Anybody software. Kinematics were measured and used as inputs. GRFs and robot actuators were simulated as virtual muscles and solved together with human muscles in an optimization process to find the robot actuator torques. 

Zhou et al. [10] simulated the interaction of an arm exoskeleton on a human subject using Anybody. The measured kinematic data were used as inputs. The assistive torques of the passive robot were pre-calculated using the kinematic data and used as inputs in the musculoskeletal model to calculate human joint torques. The robot–musculoskeletal model was wrapped into an optimization algorithm to find the optimal stiffness for the exoskeleton joints to minimize maximal muscle activity. Kim et al. [11] simulated the interactions of a hip-assist robot with a human user in Anybody for a box lifting task. The assistive torques were pre-calculated and used as inputs in the model. The weight of the robot was used in the model as an input, but its inertial effects were not simulated. A GRF prediction algorithm was used to estimate GRFs when the robot was worn. Then inverse dynamics analysis in Anybody was used to find muscle forces.

Manns et al. [12] used a different approach and simulated the dynamics of a passive lower-back robot–human system in a physics-based computational model. Optimal control and forward dynamics simulations were used to generate kinematics and to optimize robot parameters for a box lifting task.

Vantilt et al. [13] used optimization to estimate contact points and modelled human interactions with the robot as an independent disturbance acting on the exoskeleton joints. While the exoskeleton provided assistance, a desired vector of interaction forces and torques were added to the exoskeleton dynamics. Huang et al. [14] used reinforcement learning method based on policy improvement and path integrals to simulate the interactions between the robot and the user online and to estimate movement trajectories. Serrancoli et al. [15] proposed a three-phase optimal control framework to predict subject-exoskeleton collaborative movements and their interaction forces. The human and exoskeleton were each represented as a two-legged planar torque-driven model (foot, shank, thigh, and pelvis) connected together using springs and dampers. In the first step, the parameters of a smooth foot–ground Hunt-Crossley contact model with three degrees of freedom (DoFs) were identified to simulate the force between the exoskeleton and the ground. The foot–ground contact parameter values were optimized so that they could reproduce experimental contact forces. In the next step, the parameters of the spring-damper human–robot interaction (HRI) system were identified using optimization. Finally, the calibrated foot–ground and subject–exoskeleton contact parameters were used to predict sit-to-stand kinematics and HRI forces for six different trials using optimization. 

These methods provide powerful tools for early-stage design optimization. However, they (a) are computationally demanding; (b) are very prone to input errors due to the large number of uncertain input parameters and assumptions needed; (c) require measured contact forces; and (d) have outputs which are often challenging to interpret in terms of the relationship between the robot inputs and the changes in the user dynamics. 

This study proposes a computationally efficient and transparent modelling framework to simulate human-wearable robot interactions during walking. The proposed framework solves the closed-form system of equations of equilibrium of the link segment model of the human–robot system to find interaction forces and torques without optimization and walking GRF measurements. This is made possible through:(a)Defining assistive objectives of the robot in terms of unknown forces and torques in the human-robot system of equations of equilibrium to keep it statically determinate, allowing to solve it directly without optimization.(b)Using the technique developed by Shahabpoor and Pavic [16] to estimate the walking GRFs from the reference measured kinematic data, avoiding the need to measure walking GRFs.

The robot is linked directly to the human at connection points using revolute joints, and no mass–spring–damper dynamics [14,17,18] are considered at the interaction points. The proposed framework is applicable to a wide range of wearable robots whose human–robot link segment model is statically determinate. Such modeling framework offers a number of advantages:(1)Computational efficiency: the combination of a simplified link segment model and directly solving the system of equations for joint and interaction forces and torques without optimization makes the process highly computationally efficient. This, not only allows for effortless offline design optimization of the robot, but also allows the framework to be used online as part of the robot controller for real-time calculation of assistive torques.(2)Transparency: the clear mathematical formulation of the human–robot system allows for direct analysis of the relationship between the robot interventions and the consequent changes in human kinematics and kinetics.(3)Less prone to input errors due to the limited number of assumptions and uncertain inputs used.

The gait measurements carried out on six test subjects to measure the reference movement trajectories are explained in Section 2. Section 3 and Section 4 describe the proposed HRI modelling framework and its application to the lower-limb walking assist exoskeleton optimized in this study. Section 5 presents three aspects of the exoskeleton design, optimized using the proposed HRI framework, i.e., the optimal location of the robot hip and ankle joints and the actuator torque–velocity requirements. The results are discussed in Section 6 and the conclusions are highlighted.

## 2. Experimental Measurements

Six healthy male subjects, S1–S6 (age: 21 ± 1 years, weight: 77 ± 16 kg and height: 1.82 ± 0.08 m), participated in a set of walking gait measurement in the biomechanics laboratory at the University of Sheffield using an instrumented dual-belt treadmill [19]. The subjects provided informed consent in accordance with the ethical guidelines for research involving human participants at the University of Sheffield. The normal walking speed of each subject was initially found, by trial and error, to be 3 equal to $v_{w,S1}=1.25\;m/s$, $v_{w,S2}=1.28\;m/s$, $v_{w,S3}=1.28\;m/s$, $v_{w,S4}=1.11\;m/s$, $v_{w,S5}=1.19\;m/s$, and $v_{w,S6}=1.06\;m/s$, respectively. Then subjects S1–S4 each participated in a set of six walking tests with 180 s duration and treadmill speed set to 60–110% of their normal walking speed at 10% intervals, respectively. Subjects S5 and S6 each underwent a single walking test only with their natural walking speed.

In each test, the full-body 3D kinematic data were recorded using the CODA motion capture system [20] at a 100 Hz sampling rate. The marker placement protocol was based on full-body Plug-in Gait [21] (Figure 1). The tri-axial walking $GRF\left(t\right)$ signals pertinent to each foot were recorded at a 1 kHz sampling rate using the instrumented treadmill with two separate belts, each running on a six-axis force plate. The measured $GRF\left(t\right)$ signals were only used for verification and not as an input to the proposed model.

All the measured data were re-sampled at 100 Hz and synched using MATLAB software [22]. The raw kinematic data (tri-axial displacements) were filtered using a low-pass zero-lag fourth-order Butterworth digital filter with a cut off frequency of 12 Hz to remove high-frequency noise while preserving the frequency content corresponding to the first four harmonics and sub-harmonics of the walking $GRF\left(t\right)$ signals. The displacement signals were then double-differentiated to find the corresponding acceleration signals. Before each differentiation, signals were low-pass filtered using the mentioned Butterworth filter to reduce the high-frequency noise associated with the differentiation process [23]. These tri-axial acceleration signals, calculated for all CODA markers on the body, were subsequently used to calculate the acceleration of the center of mass (CoM) of each segment used in the model to estimate the walking $GRF\left(t\right)$.

## 3. HRI Modelling Framework

The HRI model proposed here is based on Inverse Dynamics (ID) analysis of the combined human–robot system. The assumptions are: (a) the target/reference human movement trajectories are known; (b) the degree of indeterminacy of the human–robot system is equal to the sum of the robot’s active and free DoFs. In another words, the link segment model of the human–robot system is statically determinate, assuming that the actuator inputs are known—a wide range of wearable robots meet this criterion (see Table 1 for five examples of such robots); and (c) there is a one-to-one relationship between the kinematics of the robot and the human. The key steps of the analysis are (Figure 2):A.Human and robot are each modelled with an appropriate link segment (LS) model with lumped masses at each segment’s CoM and linked together based on the actual HRI configuration.B.Inverse Kinematics analysis is carried out to calculate the movements of human and robot segments from the measured marker trajectories.C.The contact forces between the human–robot system and the environment are estimated/measured.D.The system of equations of equilibrium for the human–robot LS model is formulated and solved for the net joint/interaction forces and torques (ID analysis). To make the system of equations statically determinate, the robot should have the same number of passive or active DoFs as the degree of indeterminacy of the human–robot system. Torque at each passive joint is set to zero. Assistive objectives of the robot are then used to define extra force/torque constraints, equal to the number of robot’s active DoFs, to make the system statically determinate.

The proposed modelling framework is generic and can be adapted for different robot configurations, assistive targets, and movements. It provides a very clear picture of how many/which DoFs can be controlled by actuating each robot joint, and the consequent effects on human joints’ torques and forces.

## 4. Application of the HRI Framework to an Assistive Lower-Limb Exoskeleton

The proposed HRI framework is applied to a seven-segment lower-limb weight-support exoskeleton that sits medial to user’s legs and is attached to the user’s body at the hip and feet (Figure 3a). The exoskeleton is aimed at assisting the user during walking by supporting part of the user’s weight while keeping the reference movement trajectories unchanged. The analysis presented in this paper is carried out only on the sagittal plane (2D), but the methodology could be implemented in 3D. Inputs to the model are the reference walking trajectories of the human subjects measured in the experiments, subjects’ anthropometric information, and the robot’s geometry, active and passive DoFs, and segmental inertial properties. 

### 4.1. Step A

Human and robot are each modelled with an appropriate link segment model with lumped masses located at each segment’s CoM (Figure 3). The human body was modelled as an articulated eight-segment 2D system: head–arms–trunk (HAT), thighs, shanks, feet and a massless pelvis (Figure 3b). The anthropometric data for each body segment including anatomical coordinate systems, joint center definitions, the segmental masses and their CoM location are based on the system suggested by Ren et al. [24] and Winter, [25]. The pelvis dynamics were considered part of the HAT segment. To realistically represent the thigh motion, a massless virtual pelvis segment was considered to allow the hip joints to move independently (not concentric) on the sagittal plane. The HAT segment was connected to the massless pelvis segment at the midpoint of the hip joints (femoral heads).

A seven-segment (pelvis, thighs, shanks, and feet) planar model was used to simulate the dynamics of the wearable robot in the sagittal plane (Figure 3c). Six joints of the robot (hips, knees, and ankles) were revolute joints, providing six rotational DoFs that could be active or passive. The robot connected to the user only at three points: to each sole of the user’s shoes and their pelvis. This is to keep the interaction points to a minimum, the interaction forces and torques under strict control, and to make the robot robust against alignment errors [3,26]. The robot feet and pelvis segments were assumed to be fixed to the corresponding subject’s segments and massless. As a result, the robot model could be simplified to a five-segment (pelvis, thighs, and shanks) model, connected to the subject at the pelvis and ankles (Figure 3d). The robot’s co-centric revolute hip joint was located on the vertical line passing through the midpoint of subject’s femoral heads. The distance of the robot hip joint from the femoral heads midpoint, however, could be adjusted in the model, allowing for analyzing different configurations of the robot. 

### 4.2. Step B

Inverse kinematics analysis is carried out to calculate movements (displacement and acceleration) of human joints and segmental CoMs, using the measured walking trajectories in the tests as reference. The robot architecture meant that there is a one-to-one relationship between the kinematics of the robot and those of the human. Since the robot’s ankles and pelvis are attached to those of the user, their kinematics are known. The position of the robot’s knee joints were determined at each time step using circle–circle intersection, utilizing the known position of the robot’s ankle and hip joints, and the known length of robots’ thigh and shank. 

### 4.3. Step C

The walking GRFs are the only contact forces in this simulation. The anterior–posterior and vertical walking GRFs and plantar center of pressure were estimated using the kinematics of the human and the robot, based on the method proposed by Shahabpoor and Pavic [16]. The GRFs were estimated for both human-only and human–robot combined systems. The human-only walking GRFs is used as reference for defining assistive objectives in the model where needed. The Shahabpoor and Pavic [16] method also provided the toe-off and heel-strike timings which were used to specify the single- and double-support phases of each gait cycle. Any other contact forces that cannot be estimated directly using reference human movements must be measured/estimated and used as inputs to the model.

### 4.4. Steps D

Inverse dynamics analysis is then carried out on the combined human–robot link segment model, to calculate the joint forces and torques. The human–robot LS model is divided into six parts: feet, legs, pelvis, and HAT (Figure 4a). In each time-step, it is initially established if the subject is in single-support (SS) or double-support (DS) phase. In the SS phase, ID analysis began with the swing foot segment (Figure 4a—Step I), where external forces and torque are zero, followed by the swing leg (Step II), HAT (Step III), pelvis (Step IV), the stance leg (Step V), and the stance foot (Step VI) consecutively. In the DS phase, ID analysis began with the trailing foot and its estimated GRFs and plantar center of pressure (CoP) were used as the external forces to calculate ankle $F_{A,x},\;\;F_{A,z}$ and $M_{A,y}$ (Figure 4a—Step I). Subsequently, the trailing leg (Step II), HAT (Step III), pelvis (Step IV), the leading leg (Step V) and the leading foot (Step VI) were analyzed, consecutively.

Following the above sequence, for the feet, HAT, and pelvis segments, the distal joint forces and torque are known (or calculated in the previous Step) and the forces and torque at the proximal joint were calculated directly using the equilibrium of forces and torques at the segment’s CoM [24,25] (Figure 4d):(1)$$\sum F_{x}=m{\ddot{x}}_{CoM}\to F_{Px}=m{\ddot{x}}_{CoM}-F_{Dx}$$
(2)$$\sum F_{z}=m{\ddot{z}}_{CoM}\to F_{Pz}=m{\ddot{z}}_{CoM}-F_{Dz}+mg$$
(3)$$\begin{array}{c} \sum M_{y,CoM}=I_{CoM}{\ddot{\theta }}_{CoM}\to M_{Py}=I_{CoM}{\ddot{\theta }}_{CoM}-M_{Dy}-F_{Px}\left(z_{P}-z_{CoM}\right)\\ -F_{Dx}\left(z_{D}-z_{CoM}\right)-F_{Pz}\left(x_{CoM}-x_{P}\right)-F_{Dz}\left(x_{CoM}-x_{D}\right) \end{array}$$

In Equations (1)–(3), P is proximal, D is distal, m is the mass of the segment, $I_{CoM}$ is the rotational inertia of the segment around its CoM, and g is the gravitational constant.

The closed kinematic chain of the human–robot leg in Step II and V has four segments (shanks and thighs) with 18 unknown joint forces and torques (Figure 4c). In Step II, the total forces ($F_{A,x},\;F_{A,z})$ and torque $\left(M_{A,y}\right)$ at the ankle are calculated in Step I. Similarly, in Step V, the total forces ($F_{H,x},\;F_{H,z})$ and torque $\left(M_{H,y}\right)$ at the hip are calculated in Step IV. These three constraints combined with the three equations of equilibrium (1)–(3) that can be written for each segment provide 15 equations. A further three constraints are required for the system to be statically determinate. The exoskeleton has three DoFs in each leg: hip, knee and ankle rotation around the y axis. Each passive joint provides a $M_{R,y}=0$ constraint. For each active joint, a force/torque constraint defining the assistive objective of the robot needs to be defined. For instance, in the case of the weight-supporting exoskeleton simulated in Section 5.1, the knee and ankle joints are active and the hip joint is passive. Therefore, the robot hip torque was set to zero $M_{RH,y}=0$, and the remaining two constraints were chosen as $F_{RH,x}=0$ and $F_{RH,z}=c\times GRF\left(z\right)$ where (0<c<1), so that the robot applies only a vertical assistive force to the user’s hip with the magnitude $c\times GRF\left(z\right)$. These added constraints make the leg system statically determinate, and it can be solved to find the unknown joint forces and torques.

## 5. Design Optimization

Three aspects of the robot design are optimized in this study using the proposed HRI framework: position of the center of rotation (CoR) of the hip joint (Section 5.1), position of the CoR of the ankle joint (Section 5.2), and the torque–velocity requirements of the active joints (Section 5.3).

### 5.1. Center of Rotation of Hip Joint

The robot simulated in this section have four active DoFs, i.e., the knees and ankles, and the hip joints were considered passive. Three configurations of the hip joint were compared: a revolute joint with the CoR 200 mm inferior to the mid-point of the user’s femoral heads (configuration (a)); and a rail joint with the CoR at the mid-point of the user’s femoral heads (configuration (b)) and at standing CoM (configuration (c)) (Figure 5). The rail hip joint provides a better load transfer to the CoM compared to a revolute joint, but at the cost of a larger, heavier, and more complex assembly. The eccentricity of the robot hip from the user’s CoM means that the assistive forces $F_{RH,x}$ and $F_{RH,z}$ generate unwanted torque at the user’s CoM. The HRI model was used to compare this unwanted torque for the three configurations. Since the robot hip joints are passive, $M_{RH,y}$ was set to zero. To simulate the robot supporting the full weight of the user in the vertical direction without affecting the anterior–posterior motion, the assistive forces at the robot’s hip were set to $F_{RH,x}=0$ and $F_{RH,z}=GRF\left(z\right)$.

A pair of carriage-rail hip joints were designed for Configurations (b) and (c), and their geometry and inertial properties were used in the HRI model (Figure 3a). Each rail is a circular guideway forming part of the robot pelvis, carrying a sliding carriage attached to the robot thigh.

The torque profiles created by the robot at user’s CoM ($M_{R,y}\;CoM$) were calculated for all six walking subjects and overlaid for each gait cycle for comparison. Figure 6a compares $M_{R,y}\;CoM$ for configurations (a), (b), and (c) for Subject 1. The torque magnitudes are the average of the left and right legs and are normalized by the subject’s height and weight. $M_{R,y}\;CoM$ in configuration (c) is zero (benchmark configuration) since the hip CoR coincides with the CoM, and $F_{RH,x}$ and $F_{RH,z}$ do not create any torque at the CoM. The torque profiles are correlated for configurations (a) and (b).

The peak torque magnitudes are further averaged over all gait cycles and are compared in Figure 6b for all six subjects and configurations (a)–(c). As can be seen in Figure 6b, the peak torque magnitudes are significant and the magnitude is approximately an order of magnitude higher in configuration (a) compared to (b). In configuration (b), where the CoR of the robot’s hip coincide with the biological hip, although unwanted torques are exerted to the user’s pelvis, the torque profiles are correlated with the biological hip torques, resulting in a more natural user experience. 

### 5.2. Center of Rotation of Ankle Joint

To actuate the ankle, the actuator needs to be either close to the ankle where its mass creates large inertia during walking or close to the hip/torso with better inertial efficiency but with the challenge of power transmission down to the ankles. As a result, it is desirable to keep the robot’s ankle passive. In this section, the optimal position of the CoR of a passive ankle joint is analyzed and compared with an active joint. Three ankle configurations are compared with the CoR located at (a) the rear of the foot (the same position as the biological ankle), (b) the midfoot, and c) the forefoot (Figure 7). A robot with a passive rail hip joint with the CoR at the subject’s CoM and active knee joints was used in simulations.

A passive robot ankle is unable to replicate the plantar CoP trajectory of a walking human. During the stance phase, on average, the plantar CoP is located 30% of the time at the rear of the foot, 30% at the midfoot and 40% at the forefoot [27]. This means that assistive force $F_{R}\left(t\right)$ and biological force $F_{H}\left(t\right)$ at the CoM (Figure 7) have similar orientations in 30% of the stance phase in configuration (a), 30% in configuration (b), and 40% in configuration (c).

From a GRF point of view, the first peak of GRFs (weight acceptance) happens during the heel strike–foot flat phase, while the second peak (push off) happens during the heel off–toe off phase [28] (Figure 8). Therefore, only configuration (a) can effectively contribute to the first peak and configuration (c) to the second peak without creating substantial torque at the biological ankle.

The three configurations discussed above are cross-compared for both passive and active robot ankle joints in terms of the following:
(a)the magnitude of $F_{R,x}$ and $F_{R,z}$ that the robot with a passive ankle can provide at the user’s CoM over the gait cycle (Figure 9a,b);(b)the magnitude of torque at the human ankle $M_{HA,y}$ for the robot with a passive ankle (Figure 9c);(c)the ankle torque $M_{RA,y}$ required in a robot with active ankle joints (Figure 9d).

During the SS phase, $F_{R,x}$ and $F_{R,z}$ with magnitudes higher than or in the opposite direction from the corresponding biological forces $F_{H,x}$ and $F_{H,z}$ were deemed destabilising and were not allowed. During the DS phase, $F_{R,x}$ and $F_{R,z}$ for each leg were determined so as to provide 100% of $F_{H,x}$ and $F_{H,z}$. Figure 9 compares $F_{R,x}$, $F_{R,z}$, $M_{HA,y}$, and $M_{RA,y}$ for the three configurations. The graphs are the average of the curves for all gait cycles of all the subjects, normalised by their weight and height.

As can be seen in Figure 9a,b, during SS, $F_{R,x}$ is mostly the limiting factor, since assistive forces $F_{R,x}$ and $F_{R,z}$ with magnitudes higher than the corresponding biological forces ($F_{H,x}$ and $F_{H,z})$ were not allowed. For each configuration, the mean efficiency e of the assistive forces over the gait cycle is calculated as
(4)$$e=\left[\sum \left(100-\frac{\left|F_{H}-F_{R}\right|}{\max\left(F_{H}\right)}\times 100\right)\right]/100$$
and are presented in Table 2.

Further, for each torque magnitude, the absolute peak and mean values are compared in Table 3.

As can be seen in Table 2 and Table 3, configuration (c) can provide the highest $F_{R,x}$ and $F_{R,z}$ when the robot ankle is passive, but at the cost of generating higher torques at the subject’s ankle. Configuration (b), however, is the most optimal position for an active robot ankle with minimum peak torque requirement.

### 5.3. Joint Torque–Velocity Requirements

The HRI model was used to specify the actuator and gearing requirements of the robot’s active joints. The robots simulated in this section have four active DoFs, i.e., hip and knee flexion/extension, and the ankle joints were passive with CoR at the rear of the foot to avoid excessive torque at the human ankle. The assistive target was set to reduce the subject’s hip and knee flexion/extension torques by 50% while maintaining the same torque and velocity profile (100% correlated). These led to the following three added constraints for each leg model:(5)$$\left\{\begin{array}{c} M_{HH,y}=0.5\times M_{HH,y\left(unassisted\right)}\;\;\;\;\;\;\;\;\;\;\;\;\;\;\;\;\;\;\;\;\;\\ \;M_{HK,y}=0.5\times M_{HK,y\left(unassisted\right)}\;\;\;\;\;\;\;\;\;\;\;\;\;\;\;\;\;\;\;\;\;\;\\ \;\;\;M_{RA,y}=0 \end{array}\right.$$

The length of the robot’s shank and thigh were both considered 0.4 m. Maxon EC90 flat motor with nominal torque of 1.5 Nm at 1540 rpm, and a stall torque of 13.3 Nm was used in the analysis as the actuator [29]. Hip motors were located on the pelvis segment and knee motors on the thigh segments. The mass of the robot’s hip, thigh, and shank were assumed to be 3 kg, 1.5 kg, and 0.5 kg, respectively. The CoR of the robot’s revolute hip joint was considered 200 mm below the mid-point of subject’s femur necks. Figure 10 and Figure 11 show the required robot hip and knee joint torque, velocity, and power profiles, respectively, for Subject 1. The graphs are averaged over the gait cycle. 

To reduce the required gear ratio, it was deemed acceptable for the motor to go beyond nominal torque for part of the gait cycle. A typical cumulative distribution functions (CDF) of the required torque magnitudes for Subject 1 are plotted in Figure 10b and Figure 11b against gear ratios. For each gear ratio, the vertical axis on the CDF graphs shows the proportion of the gait cycle that the motor torque stays below the nominal torque. 

A low gear ratio was desirable for this robot to minimize impedance and maximize back drivability. A gear ratio of 10 was chosen to provide a relatively low gear ratio while the motor torque remained at 80% and 55% of the gait cycle below the nominal torque for the hip and knee joints, respectively. 

## 6. Discussion and Conclusions

The simple, computationally efficient and transparent closed-form solution of the human–robot interactions in the proposed method offers invaluable insight and flexibility to the designer. Since the HRI model calculates the interaction forces at the interface of the human body and the robot, the assistive objectives can be defined to directly optimize/minimize these interaction forces. The computational efficiency of the proposed HRI model further allows the model to be used online as part of the robot controller to calculate the required assistive torques in real time.

The HRI framework, however, utilizes a deterministic system of equations to avoid optimization. This entails the same number of assistive objectives as the number robot active DoFs. It further requires a one-to-one relationship between the user and robot’s kinematics. The model requires the reference movement trajectories and the contact forces as input. As in ID analysis, the joint forces and torques calculated using this method are the ‘net’ forces and torques and the counterbalancing actions due to co-contraction of contralateral muscles cannot be calculated.

Finally, although kinematic targets cannot be used explicitly in the HRI model as assistive objectives, any desired movement can be measured and used as an input to the model, which implicitly acts as a kinematic objective.

## Figures and Tables

**Figure 1 sensors-24-04081-f001:**
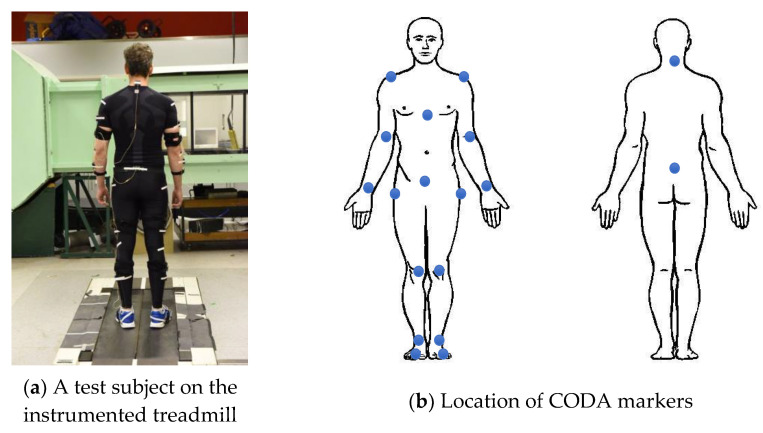
Test subject instrumentation.

**Figure 2 sensors-24-04081-f002:**
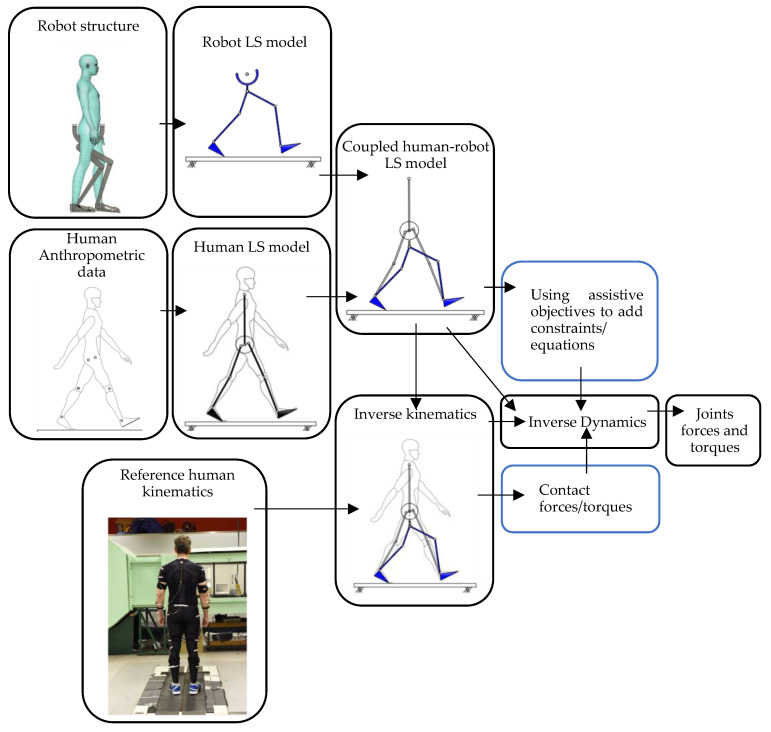
HRI modelling framework.

**Figure 3 sensors-24-04081-f003:**
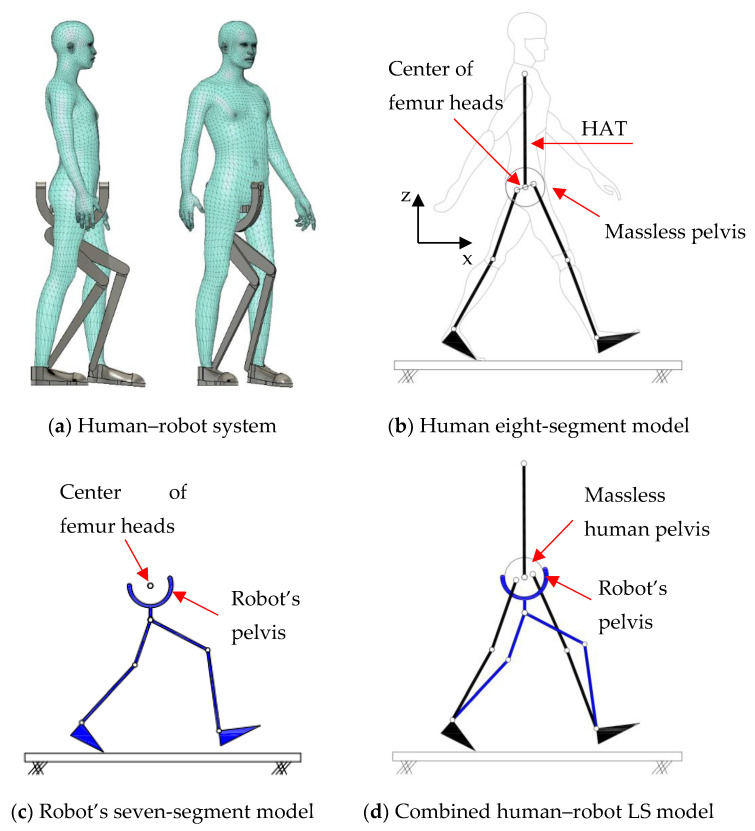
Link segment model of the human–robot system.

**Figure 4 sensors-24-04081-f004:**
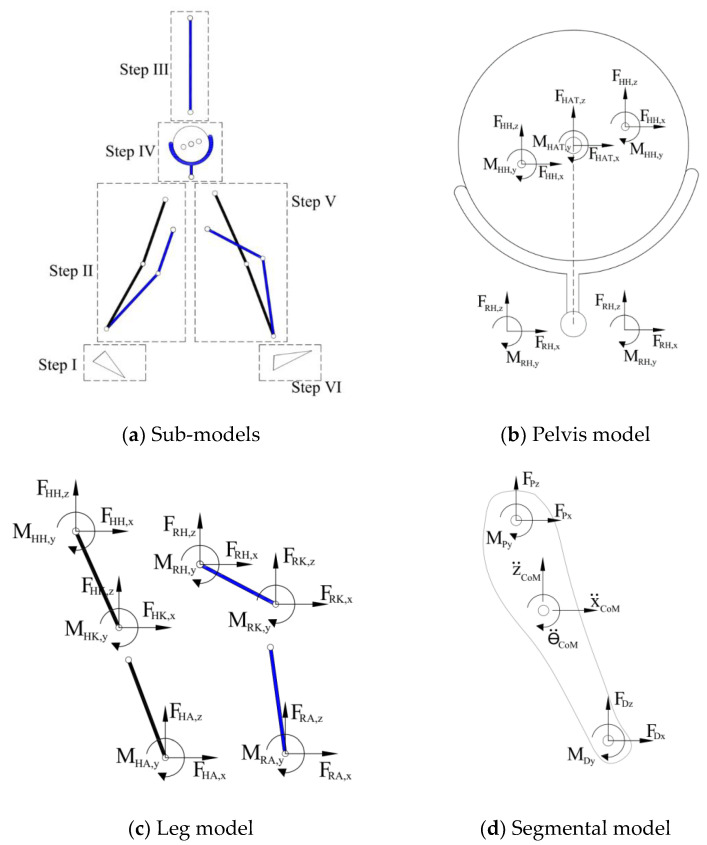
ID analysis of link segment planar free-body diagrams.

**Figure 5 sensors-24-04081-f005:**
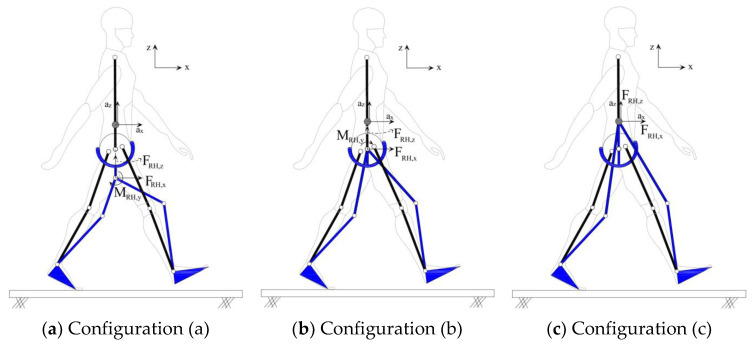
Hip joint configurations.

**Figure 6 sensors-24-04081-f006:**
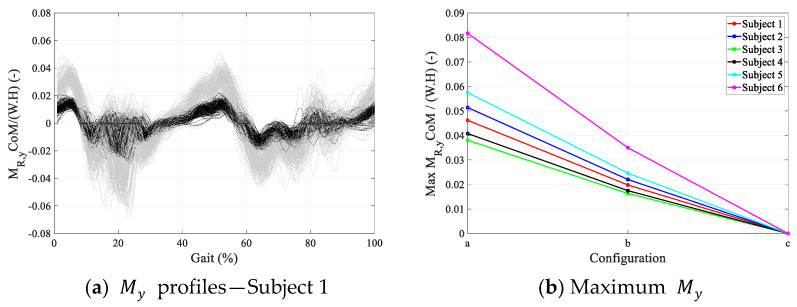
$M_{R,y}$ at user’s CoM: gray: configuration (a); black: configuration (b); $M_{R,y}$ in configuration (c) is zero.

**Figure 7 sensors-24-04081-f007:**
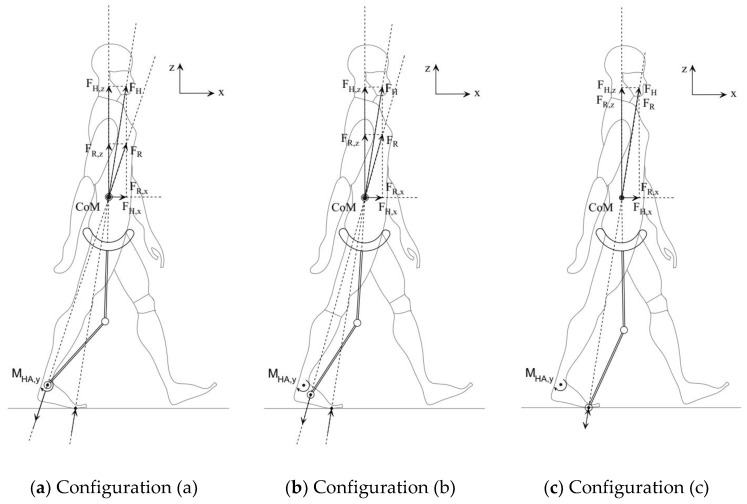
Position of robot’s ankle with respect to the human foot.

**Figure 8 sensors-24-04081-f008:**
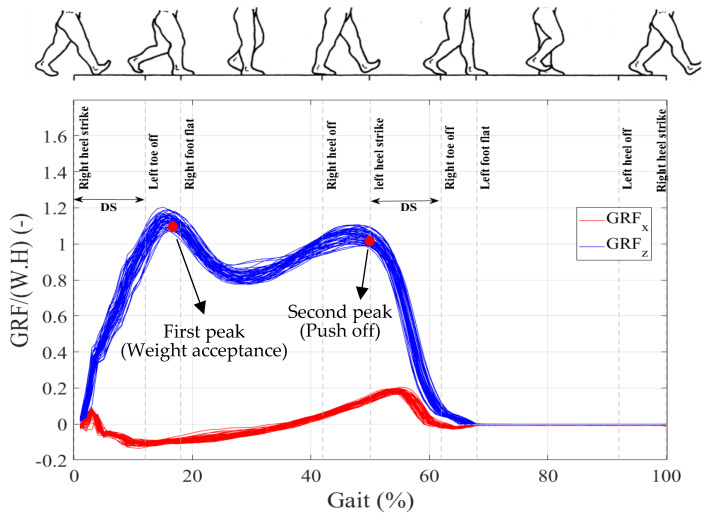
Walking GRF(t)s of the right leg mapped to the timing of the gait.

**Figure 9 sensors-24-04081-f009:**
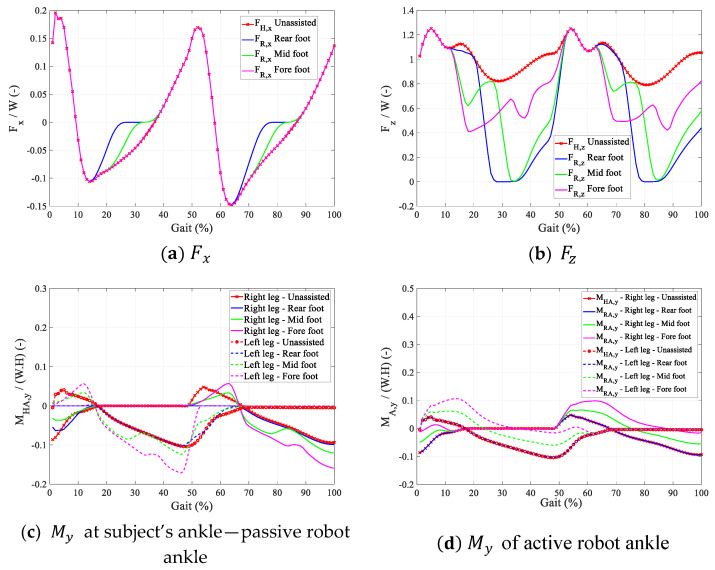
Comparison of assistive and biological ankle forces and torques.

**Figure 10 sensors-24-04081-f010:**
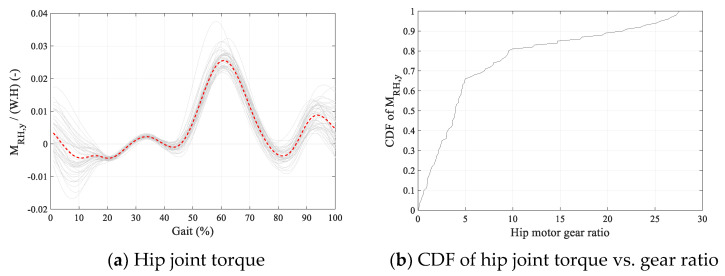
Robot hip joint torque–velocity requirements for Subject 1. The average is shown in dashed red in (**a**,**c**).

**Figure 11 sensors-24-04081-f011:**
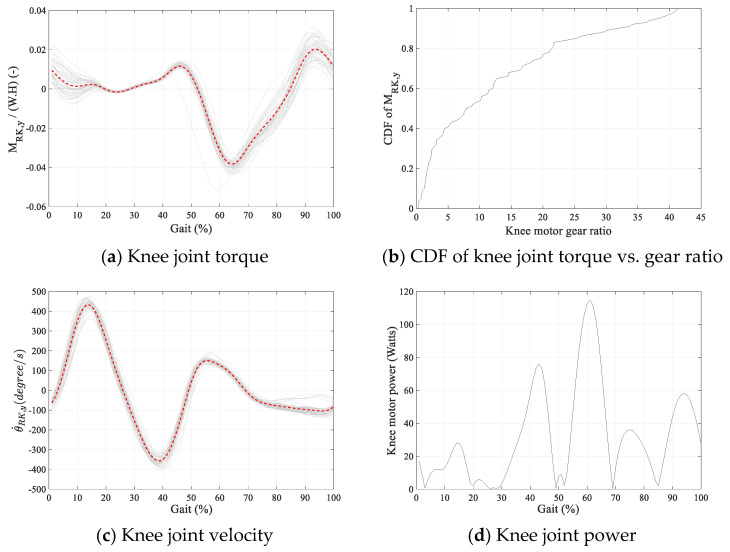
Knee joint torque–velocity requirements for Subject 1. The average is shown in dashed red in (**a**,**c**).

**Table 1 sensors-24-04081-t001:** Examples of statically determinate wearable robots.

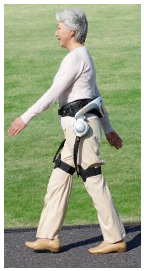	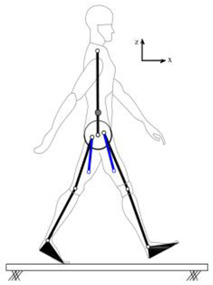	Honda walking assist: 1 active DoF per leg: hip 15 unknowns and 15 equations and constraints In each leg link segment model: 3 unknowns per joint ($F_{x},\;F_{z}$ and $M_{y}$): 15 total unknowns 3 equilibrium equations per segment: 9 equations 3 known ground reactions, i.e., known ankle total $F_{A,x},\;F_{A,z}$ and $M_{A,y}$ 3 known constraints: ○ The robot hip torque $M_{RH,y}$ is generated by the actuator. ○ The axial force on the robot thigh segment is zero. ○ The torque at connection of the robot with the user’s thigh is zero.
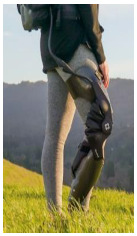	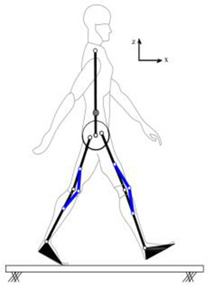	Ascend robotic knee brace (Roam Robotics): 1 active DoF per leg: knee 18 unknowns and 18 equations and constraints In each leg link segment model: 3 unknowns per joint ($F_{x},\;F_{z}$ and $M_{y})$: 18 total unknowns 3 equilibrium equations per segment: 12 equations 3 known ground reactions, i.e., known ankle total $F_{A,x},\;F_{A,z}$ and $M_{A,y}$ 3 known constraints: ○ The robot knee torque $M_{RK,y}$ is generated by the actuator. ○ The torque at connection of the robot with the user’s thigh is zero. ○ The torque at connection of the robot with the user’s shank is zero.
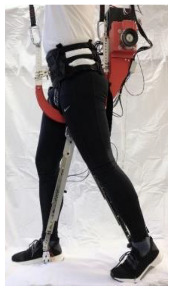	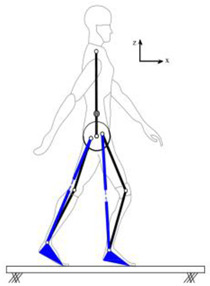	Fall prevention robot developed by the authors: 1 active DoF per leg: hip 15 unknowns and 15 equations and constraints In each leg link segment model: 3 unknowns per joint ($F_{x},\;F_{z}$ and $M_{y}$): 15 total unknowns 3 equilibrium equations per segment: 9 equations 3 known ground reactions, i.e., known ankle total $F_{A,x},\;F_{A,z}$ and $M_{A,y}$ 3 known constraints: ○ The robot hip torque $M_{RH,y}$ is generated by the actuator. ○ The axial force in the robot thigh is zero (passive telescopic leg). ○ The robot ankle torque $M_{RA,y}$ is zero (passive revolute joint).
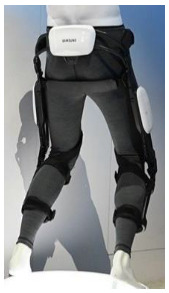	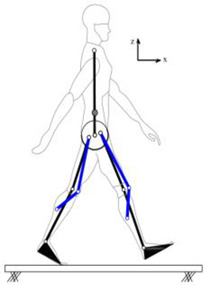	Samsung walking assist robot: 2 active DoFs per leg: hip and knee 18 unknowns and 18 equations and constraints In each leg link segment model: 3 unknowns per joint ($F_{x},\;F_{z}$ and $M_{y})$: 18 total unknowns 3 equilibrium equations per segment: 12 equations 3 known ground reactions,, i.e., known ankle total $F_{A,x},\;F_{A,z}$ and $M_{A,y}$ 3 known constraints: ○ The robot hip torque $M_{RH,y}$ is generated by the actuator. ○ The robot knee torque $M_{RK,y}$ is generated by the actuator. ○ The torque at connection of the robot with the user’s shank is zero.
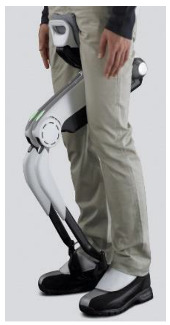	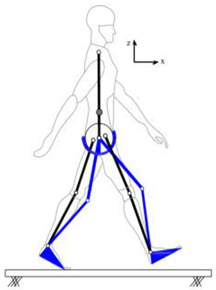	Honda weight support robot: 1 active DoFs per leg: knee 18 unknowns and 18 equations and constraints In each leg link segment model: 3 unknowns per joint ($F_{x},\;F_{z}$ and $M_{y})$: 18 total unknowns 3 equilibrium equations per segment: 12 equations 3 known ground reactions, i.e., known ankle total $F_{A,x},\;F_{A,z}$ and $M_{A,y}$ 3 known constraints: ○ The robot knee torque $M_{RK,y}$ is generated by the actuator. ○ The robot hip torque $M_{RH,y}$ is zero (passive revolute joint). ○ The robot ankle torque $M_{RA,y}$ is zero (passive revolute joint).

**Table 2 sensors-24-04081-t002:** Assistance efficiency.

	$F_{R,x}$ Conf (a)	$F_{R,x}$ Conf (b)	$F_{R,x}$ Conf (c)	$F_{R,z}$ Conf (a)	$F_{R,z}$ Conf (b)	$F_{R,z}$ Conf (c)
e	92%	97%	99%	68%	73%	80%

**Table 3 sensors-24-04081-t003:** Mean and peak ankle torques.

	$M_{HA,y}$ Unassisted	$M_{HA,y}$ Conf (a)	$M_{HA,y}$ Conf (b)	$M_{HA,y}$ Conf (c)	$M_{RA,y}$ Conf (a)	$M_{RA,y}$ Conf (b)	$M_{RA,y}$ Conf (c)
$\max\left(\left|M_{y}\right|\right)$	0.09	0.10	0.12	0.16	0.09	0.07	0.10
∑My100	0.028	0.025	0.032	0.038	0.028	0.022	0.023

## Data Availability

Data are contained within the article.

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
