# Peer review of "Wearable Robot Design Optimization Using Closed-Form Human–Robot Dynamic Interaction Model"

_sensors, 2024, doi:10.3390/s24134081_

Round 1
Reviewer 1 Report
Comments and Suggestions for Authors
This paper proposed a framework to simulate mathematically the human-robot interactions based on a simplified link-segment model of the human subject and the robot. The topic is interesting in the fields of sensing and robotic control. However, the contributions of this paper need further clarification. Some comments are given to improve the paper as follows:
1) The motivation of this study is not clear enough. Some recent references should be compared and the differences with the existing related references should be discussed. In fact, many methods were proposed to consider for HRI modeling. Additionally, the authors claimed that no optimization or contact forces sensing was needed, which seems somewhat exaggerated.
2) More details should be provided for robot modeling. Specifically, the authors said that the human-robot interaction model was built by using the inverse dynamics approach. However, it is hard to see how to use the inverse dynamics approach.
3) Some important factors were ignored in the robot modeling. For example, joint actuators was not considered. The authors can be refereed to the related work https://doi.org/10.3390/act12040147.
4) The paper was not well written. Many figures are not clear. There are many grammar mistakes. For example, (Error! Reference source not found), and in Figure 8, the symbol c is incorrect. The expression of curve b in Figure 11 is unclear.
5) The literature review was not sufficient. Many recent results are not reported. For example, many data-based impedance controls have been studied for HRI modeling, such as DOI: 10.1109/TASE.2018.2886376.
Comments on the Quality of English Language
The quality of English language needs to be further improved.
Reviewer 2 Report
Comments and Suggestions for Authors
This paper discusses wearable robots for assisting individuals with mobility disorders. It emphasizes the use of virtual prototyping to design these robots efficiently, avoiding costly physical iterations. The paper proposes a framework based on a simplified formulation, allowing for simulation without optimization or intricate models. This framework is applied to optimize the design of a lower-limb wearable robot for seven segments with six degrees of freedom, focusing on the hip and ankle joints and actuator requirements. According to the authors, Gait measurements from healthy subjects validate the design.
In general, the paper is not well written or explained. The mathematical level to indicate the kinematic and dynamic models of agents and robots is almost non-existent. I have the following recommendations.
- The paper is very poorly organized. Figure 1 appears 3 times within the article. Additionally, there are references to equations that do not exist, and the error message is displayed in the text. There is a section called Section 0, and it should be 1. Carelessness on the part of the authors can be observed in the writing and design of the paper.
- In the methodology part, it is necessary to indicate in a general architecture such as block diagrams how everything that is presented in the article is connected.
- There are parts of the text where kinematic models are talked about, but there is no reference or equation that indicates these models. Below I leave some sections of text where this problem occurs.
"Since the human and robot models are kinematically coupled together, inverse kinematics analysis was used to calculate robot's movement based on the known human subject's movement (measured experimentally here). Here, a one-to-one relationship is assumed between the kinematics of the robot and the human. If the robot is kinematically redundant, appropriate motion planning/optimization techniques must be integrated into the model to select the desired motion trajectory set for the robot.", "A coupled rigid link-segment planar model was developed to simulate the interactions of the human and robot during walking. The analysis presented in this paper is carried out only in the sagittal plane (2D), but the methodology could easily be implemented in 3D." and "A 7-segment (pelvis, thighs, shanks and feet) planar model was used to simulate the
dynamics of the wearable robot in the sagittal plane (Error! Reference source not found.b)."
- The authors must consider that to control a robot, they must have measurements (which is what the authors obtain in the paper), the mathematical model of the system (kinematic or dynamic), and the design of the controller, which tries to reduce the error between measurements and reference signals that are necessary for the robot to perform a task with a certain level of autonomy. However, there is no information on how they model the human and the robot. It is very difficult to follow the reading of this article, which has practically no equations where they develop the model of the robot and the person. This is a serious problem since the paper deals with the construction of a "Human-Robot Dynamic Interaction Model", and I don't see that model anywhere. What has been done are measurements, and estimates based on them. Nor is any method used to estimate mathematical model parameters based on the measurements made, which is perhaps one of the ways to solve the problem they present.
- A table is necessary with the meanings of abbreviations and symbols of the variables used in the mathematical approach.
In summary, the paper is not easy to follow, and the writing is incomplete and not detailed. There are not enough references, there are repeated figures, and above all, I cannot understand what Human-Robot Dynamic Interaction Model the authors propose, although clearly, there are practically no definitions or mathematical models explained or developed in the article. I suggest the authors continue improving and working on this topic, which can be very useful in the future of human-robot interaction. However, in this state, I do not recommend this paper for publication.
Round 2
Reviewer 1 Report
Comments and Suggestions for Authors
The revision still did not give sufficient discussions on the existing works https://doi.org/10.3390/act12040147, https://doi.org/10.1016/j.neucom.2023.126963, doi:10.1109/TASE.2018.2886376. Please explain the differences between the existing methods and the work in the revised manuscript to show the motivation.
Comments on the Quality of English LanguageSome grammar mistakes still exist.
Reviewer 2 Report
Comments and Suggestions for Authors
This paper discusses wearable robots. It emphasizes the use of virtual prototyping to design these robots efficiently, avoiding costly or complicated physical models. This framework is proposed to optimize the design of a lower-limb wearable robot for seven segments with six degrees of freedom, focusing on the hip and ankle joints and actuator requirements. According to the authors, Gait measurements from healthy subjects validate the design.
The authors have done important work trying to improve this article. However, it still is not well written or explained. The mathematical level to indicate the kinematic and dynamic models is still really hard to understand and it is not validated. The authors also assume a lot of things without proper justifications. I have the following comments and recommendations.
1) Again, I appreciate the effort of the authors of this article to try to improve the understanding of the paper. However, despite the authors' efforts, I have a hard time understanding the article. The writing and structure of this article have improved, but it does not allow me to understand what was done in this paper. I have to read each section more than 3 times and still is hard to get. The results figures are clear, but the formulation of the paper remains confusing. Above all, the question remains as to how good this model is, compared to methods that are normally used for topics related to dynamic modeling of solids in 3D space, such as, for example, the Euler-Lagrange method.
During development, various forces and torques are said to be zero, but it is not well explained why. Several things are assumed within the methodology of the article, but they are not justified properly.
There is no ground truth to evaluate how good this model is concerning more formal models that study the dynamics of rigid bodies, for example, Eurler-Lagrange. If forces and torques are estimated in any way, the estimation error must be calculated to properly evaluate the proposed model. The article seems more like a paper in which human data is obtained, and based on several assumptions (which are not explained very well), it is about estimating forces and torques, and the estimation error is not calculated, so It is not known how good the model is. Likewise, analysis of torque, speeds, and motor power is done in the figures in the results parts, but it is not well explained how this is modeled.
The paper remains difficult to read. A modeling paper without equations that explain in detail what is being done is hard to follow. Many things are assumed without being clear about the how, when, and why of each assumption.
I recommend to the authors that in a single architecture figure, try to explain everything that is done within the article. I also recommend the following stages for the architecture: 1) data acquisition, 2) pre-processing, 3) Formal modeling explained in very detail, 4) Model validation (how good is the model concerning other more formal models, for example, Euler-Lagrange, or other. Simulations must be presented, or the architecture of the implementation carried out with motors must be demonstrated to validate what has been exposed related to the analysis of torque, speeds, and power of motors. If there is any simulation for this, the framework of the simulation should be explained in detail.
The text in figures 3.b, 3.c, and 3.d looks incomplete in terms of the text.
From my point of view, this article is still not well explained to be considered a journal paper. Even for a conference paper, this should be better explained. I thank the authors again for trying to improve the article, and I encourage the authors to continue their journey for this interesting topic. However, I consider that this article should be rewritten and explained much better with more detail and formal analysis, especially if it is a mathematical modeling and estimation paper, it should demonstrate how good the estimated model is in comparison with more formal methods.
Comments on the Quality of English LanguagePlease, ask a collaborator to read the article, and then ask what understand of it (including the English grammar, which is sometimes confusing). Keep improving this paper with their suggestions. This topic is interesting, and I encourage the authors to keep investigating on it.!
Round 3
Reviewer 2 Report
Comments and Suggestions for Authors
The authors have responded to the comments made in the previous review successfully. In particular, improvements have been made in the following points:
- The introduction, methodology, and conclusions have been improved and revised, taking into account all the recommendations made.
- The proposed architecture, along with the details of the database used for this application have been reviewed, and are better explained.
- The quality of the figures, and their explanation has improved significantly.
- Fixed writing errors in the article.
-The authors added equations to support their formulations, which was key for this article.
- The conclusions have been improved, and the contribution of the article is explained in a better way.
Based on this, I can say that this paper can be published in its current state.